# Developmental Transformation and Reduction of Connective Cavities within the Subchondral Bone

**DOI:** 10.3390/ijms20030770

**Published:** 2019-02-12

**Authors:** Shahed Taheri, Thomas Winkler, Lia Sabrina Schenk, Carl Neuerburg, Sebastian Felix Baumbach, Jozef Zustin, Wolfgang Lehmann, Arndt F. Schilling

**Affiliations:** 1Clinic for Trauma Surgery, Orthopaedic Surgery and Plastic Surgery, Universitätsmedizin Göttingen, 37075 Göttingen, Germany; shahed.taheri@med.uni-goettingen.de (S.T.); wolfgang.lehmann@med.uni-goettingen.de (W.L.); 2Institute of Biomechanics, Technische Universität Hamburg-Harburg, 21073 Hamburg, Germany; thomas.winkler01@gmail.com (T.W.); l.schenk@implantcast.de (L.S.S.); 3Experimental Surgery and Regenerative Medicine, Department of Trauma Surgery–Campus Innenstadt, Munich University Hospital LMU, 80336 Munich, Germany; carl.neuerburg@med.uni-muenchen.de (C.N.); sebastian.baumbach@med.uni-muenchen.de (S.F.B.); 4Pathology, Universitätsklinikum Hamburg Eppendorf, Hamburg 20251, Germany; zustin@pathologie-hamburg.de

**Keywords:** bone histomorphometry, subchondral bone, microcomputed tomography, osteoarthritis, calcified cartilage

## Abstract

It is widely accepted that the subchondral bone (SCB) plays a crucial role in the physiopathology of osteoarthritis (OA), although its contribution is still debated. Much of the pre-clinical research on the role of SCB is concentrated on comparative evaluations of healthy vs. early OA or early OA vs. advanced OA cases, while neglecting how pure maturation could change the SCB’s microstructure. To assess the transformations of the healthy SCB from young age to early adulthood, we examined the microstructure and material composition of the medial condyle of the femur in calves (three months) and cattle (18 months) for the calcified cartilage (CC) and the subchondral bone plate (SCBP). The entire subchondral zone (SCZ) was significantly thicker in cattle compared to calves, although the proportion of the CC and SCBP thicknesses were relatively constant. The trabecular number (Tb.N.) and the connectivity density (Conn.D) were significantly higher in the deeper region of the SCZ, while the bone volume fraction (BV/TV), and the degree of anisotropy (DA) were more affected by age rather than the region. The mineralization increased within the first 250 µm of the SCZ irrespective of sample type, and became stable thereafter. Cattle exhibited higher mineralization than calves at all depths, with a mean Ca/P ratio of 1.59 and 1.64 for calves and cattle, respectively. Collectively, these results indicate that the SCZ is highly dynamic at early age, and CC is the most dynamic layer of the SCZ.

## 1. Introduction

Osteoarthritis (OA) is a degenerative joint disease with several verified risk factors including sex, immobility, obesity, and joint injuries [1]. Other common causes include inappropriate mechanical stress to the joint via malposition of the axis, mechanical overload, or inflammatory joint processes [2].

Although sometimes overlooked, the strongest risk factor for OA is age. Below the age of 30, OA is virtually absent [3], suggesting a change in joint physiology after young adulthood that enables OA to develop. This change could directly affect cartilage [4]. Cartilage degradation can trigger a vicious circle that eventually leads to the destruction of the joint [5]. 

Increasing evidence, however, points out that the first changes in OA are usually seen in the subchondral bone, suggesting an important early role for the subchondral bony zone (SCZ) in early OA development [6,7]. The SCZ consists of two different mineralized layers that form a single unit, separating the articular cartilage from the bone marrow [8]. The calcified cartilage (CC) is a band of mineralized matrix of collagen fibrils, proteoglycans and hydroxyapatite, which is separated from the articular cartilage by a border known as the tidemark [9]. Calcified cartilage is connected to the underlying subchondral bone plate (SCBP), a thin cortical lamella that collectively form the osteochondral junction [8]. The SCZ is known to have a relatively high number of arterial and venous vessels, suggesting a high level of metabolism, possibly through bone remodelling at this site [10]. 

Nevertheless, the majority of the OA research has been dedicated to healthy vs. early OA or early OA vs. advanced OA scenarios [11,12,13,14,15]. Few studies have focused on the microstructures of the normal healthy subchondral bone and how they change with age [9,16]. This is important since it may provide a foundation to distinguish between maturation-based microstructural changes and those of early OA, which are shown to have some overlaps [12,17].

To uncover possible microstructural changes of the SCZ at a potentially interesting time frame at the border between early age and young adulthood, we analyzed the chemical composition, histology, and bone microstructure of healthy subchondral bone in 3-month-old calves and 18-month-old cattle. Even though inherent differences between the ageing models of bovine and human might exist, we will discuss that the presented model could be valuable in addressing key microstructural changes in the maturation phase of the subchondral bone.

## 2. Results

### 2.1. Layer Thickness

Bone cylinders extracted from the medial condyle of femur (Figure 1A–D) were analyzed histomorphometrically (Figure 1E,F). The mean thickness of the entire subchondral zone was significantly higher in cattle compared to calves (calves: 494.4 ± 135.4 µm; cattle: 869.5 ± 244.6 µm; *p* < 0.01). However, the ratio of the CC thickness and the SCBP thickness was relatively unchanged. In calves, the thickness of the CC (appears as a light blue layer in Figure 1E) was 168.6 µm ± 63.7 µm, which accounts to 34.1% of the SCZ. The CC was significantly thicker (Figure 1F; 265.2 ± 84.4 µm; *p* < 0.01) in cattle, but the CC/SCZ ratio was more or less the same (30.5%). Likewise, the ratio of SCBP/SCZ was 66.3% and 69.5% for calves and cattle, respectively. Calves demonstrated a fine trabecular structure, large co-continuous trabecular cavities, and small conduits within the CC, whereas cattle were characterized by thicker trabeculae, a more compact bone structure, and only a few conduits in the CC region.

### 2.2. Transsectional Micro-CT-Analysis of the Subchondral Trabecular Bone

The three-dimensional microstructure was analysed by micro-CT (micro-computed tomography) imaging (Figure 1G,H) and properties of the subchondral trabecular bone (subarticular spongiosa) are summarized in Table 1. The mean bone volume density (BV/TV; *p* < 0.05) and trabecular thickness (Tb.Th.; *p* < 0.05) were increased with age. This was accompanied by a smaller spacing between the bone cavities (*p* < 0.05), an age-related decline for the trabecular number (Tb.N.; *p* < 0.05), and a decrease in the connectivity density (Conn.D.; *p* < 0.05). There was no statistically significant change in geometric anisotropy with age.

### 2.3. Microarchitecture

The hypothesis that the samples came from identical sources, and the bone parameters had the same distribution of scores was rejected for all morphometric indices except Tb.Th. (*p* * values). One-way ANOVA was then performed to see whether age (sample type) or depth is the predominating factor for each index. BV/TV, Tb.Sp., and DA were affected by age, while Tb.N. and Conn.D. were significantly influenced by depth. BV/TV (*p_age_* = 0.0002) and DA (*p_age_* < 0.05) were significantly higher in cattle, while Tb.Sp. showed an age-related decrease (*p_age_* < 0.01). Depth-related analysis of the individual layers of the SCZ (i.e., CC and SCBP; Table 2) revealed that Tb.N. (*p_depth_* = 0.001) and Conn.D. (*p_depth_* < 0.05) have significant depth-related decreases as the region shifts from CC to SCBP.

### 2.4. Microchannels

Within the integrated region of the SCBP and CC, there was a continuity to the conduits in a way that the intersection of cartilage and the most superior surface of the subchondral bone was directly connected to deeper trabecular bone via a series of microchannel structures. This is not to be confused with the cavity structure in the subarticular spongiosa, which is the typical trabecular spacing. The 3D-reconstructed positive images (Figure 2A,B) showed that many of the conduits, indeed, reach to the surface of the subchondral bone (marked by white arrows). For visual representation of the porosity structure within the subchondral zone, inverted 3D-images were generated, in which the red areas (Figure 2C,D) are composed of the non-osseous tissues, while the bone tissue is transparent. The microchannels were narrower and more densely-packed in calves, while thicker and less frequent in cattle. 

### 2.5. Mineralization

Older animals (mg HA/cm^3^; Figure 3) showed higher values of mineralization throughout the whole SCZ. In the first 50–250 µm (roughly corresponding to the CC region), there was a steep increase in mineralization, followed by a plateau in the SCBP.

### 2.6. Chemical Element Analysis

The “range” of changes in the values of carbon (C), calcium (Ca), and phosphorous (P) contents were narrow (Figure 4). Nevertheless, the differences between the age groups were (for the most part) statistically significant. The correlation between depth and bone chemical composition was calculated for two seemingly different regions (0–300 µm and 300–700 µm; detailed data presented in Appendix A).

Consistent with the mineralization data, the carbon (C) content was higher in calves compared to cattle at all depths, while calcium (Ca) and phosphorous (P) were lower. Comparison of the carbon–calcium (C/Ca) concentration showed a significantly higher ratio in calves compared to cattle at all depths. In calves, the highest C/Ca ratio was observed at the initial depth of 0–50 µm. The value decreased (10%) within the depth-range of 50–150 µm, and remained relatively constant at further depths. C/P ratio revealed a similar pattern. The calcium to phosphorus ratio was higher in cattle (Figure 4F), as was the case for individual age-related changes in Ca and P contents.

## 3. Discussion

In the present study, we evaluated the transformation of the subchondral zone from young age to early adulthood on a layer-by-layer basis, with the aid of microcomputed tomography, bone histomorphometry, and chemical element analysis. We found that the number of trabeculae and their connectivity increases as the region shifts from calcified cartilage to the subchondral bone plate. BV/TV, Tb.Th., and DA were more influenced by age than depth. A series of intricate microchannel structures connected the subchondral trabecular bone to the tidemark, which were more frequent and smaller in size in calves compared to cattle. Calcified cartilage demonstrated a steep mineralization profile and a chemical composition gradient.

There might be obvious interspecies differences between the bone maturation of bovine and human bones, even though comparative genomics mapping of cows and humans shows that over 80% of the examined cattle genes had human orthologs and the two genomes had at least 105 conserved chromosomal segments in common [18]. Nevertheless, the weight of 18-month-old cattle (~150 kg) is 2.5 times higher than that of calves (~57 kg) [19,20]. As joint-compressive load is considered a major contributing factor in shaping bone architecture [21], this might raise the question of whether the bovine ageing model can be compared to ageing in humans. However, in order to determine the proper correspondence between different age groups of cattle and human, the lifespan differences need to be addressed as well. Cattle have a natural lifespan of 15–23 years [22,23], while the natural lifespan of human is reportedly 75–80 years [24]. Hence, three-month-old calves and 18-month-old cattle, which are representative of early age and young adulthood are equivalent to 3–6-year-old preschoolers and 16–18-year-old young adults in humans, respectively [18,22]. According to Vorwerg et al., the weight of 3–6-year-old preschoolers is in the range of 17.3–23.0 kg, with a mean value of ~20 kg [25]. On the other hand, 16–18-year-old young adults weigh 60.8–67 kg [26,27], which accounts for ~300% weight difference between the two groups. Taking into consideration the 250% weight difference between three-month-old calves and 18-month-old cattle, it can be concluded that our sampling allows comparable maturation-based observations. Therefore, two pre-OA groups that represent young age and early adulthood were selected.

We found an expected age-related increase in the thickness of the subchondral zone that can most probably be attributed to the growth of the animals. Indeed, the ratios of the CC and SCBP were relatively constant (CC/SCZ ≃ 0.32, SCBP/SCZ ≃ 0.68) in the two sample types. Likewise, Matrinelli et al. found that the mean thickness of CC increased by over 100 µm from healthy young horses (<5 years) to horses aged 15–19 years [9]. SCZ thickness increases have been reported also in samples susceptible to early OA (e.g., cartilage lesion). Huang et al. found thicker subchondral bone in older C77B2/6 mice after destabilization of the medial meniscus (DMM) surgery [11], while increased thickness of the SCZ in cores taken from lesioned areas of human joints has been reported as well [12]. Ding et al. reported a significantly thicker medial cancellous bone in early OA, even before signs of cartilage damage [28]. If subchondral bone changes have a pathophysiological role in the development of early OA, the lower subchondral bone thickness in young joints may be protective and this may explain the low incidence of OA in the young.

The bone volume fraction of each layer of the SCZ was increased with age. This is in line with the age-related changes of the subchondral bone in healthy Labrador Retrievers [16], and DH guinea pigs [13]. Again, the differences in the BV/TV of early OA patients mirror those of healthy age-related scenarios, as indicated by a near unanimity about the increase of BV/TV in early OA [12,14,15,17]. In a previous study on 56 proximal tibias, however, Chen et al. showed age-related declines in the BV/TV [29]. This might have been due to the selected cohort in the age range of 57–98 years, which are typically in the stage of osseous deterioration. [30,31].

We found that “age” and “distance from the tidemark” have opposite influences on the trabecular spacing. Tb.Sp. decreased significantly with age (62.5%, *p* = 0.005), and increased insignificantly with distance from the tidemark (68.4%, *p* = 0.15). Moreover, connectivity—the degree to which a structure is multiply connected—showed insignificant (50%, *p* = 0.13), and significant (63%, *p* < 0.05) decreases with age and distance from the tidemark, respectively. Similar age-related observations are reported for the human femoral neck [29] and the subchondral bone of proximal tibia [32]. The decreased connectivity, coupled with increases in BV/TV and Tb.Th., suggests a remodeling mechanism in which cavities are filled via a union of trabecular bone [17,28]. 

A handful of studies have discussed the presence of a channel-like microstructure in subchondral bone that might provide a direct link between articular cartilage and subchondral trabecular bone [33,34]. Here, we illustrate that these microchannels are more abundant and narrower in calves compared to cattle (Figure 2). The fact that the BV/TV in the subchondral zone is significantly higher in cattle suggests that the total surface area of microchannels is lower in cattle. This might be important if the microchannels serve as pathways for nutritional or oxygen supply, as alluded in other reports [35], and thereby possibly making the cartilage less vulnerable against supply deficits. It may be interesting to analyze changes in this microchannel structure in correlation with cartilage thickness and other hallmark features of OA.

Among the rare studies on the chemical composition analysis of the subchondral bone is the work of Li et al. [36]. They showed that the ratio of Ca/P in the SCBP of human femoral head had no significant difference between OA, osteoporotic (OP), and control samples. In the present study, the EDX analysis of the subchondral bone was performed with a step size as low as 50 µm. Interestingly, the most perceptible depth-dependent behavior of the chemical compositions was observed within the first 150 µm from the tidemark. Considering the thickness of SCZ’s individual layers (which was evaluated histomorphometrically), it is safe to infer that this region is entirely situated within the calcified cartilage. This leads us to hypothesize that CC is more chemically dynamic compared to deeper regions of the SCBP (i.e., subchondral bone plate, and deeper subarticular spongiosa), and might play a crucial role in bone growth and developmental transformation of the SCBP. Indeed, calcified cartilage is known for its active remodeling process through endochondral ossification [37]. Likewise, the deterioration of articular cartilage is partially attributed to a recurring process in CC in which the tidemark advances towards cartilage, which may be a decisive factor in the development of OA [38]. 

The Ca/P ratio has been used as a biomarker for the evaluation of bone health, as well as tracking age-related changes [39,40]. We found mean values of Ca/P ~1.59 for calves and ~1.64 for cattle, which is consistent with a previous report on young (1–3 months; 1.51) and adult (4–5 years; 1.58) bovine cancellous bone [41]. Interestingly, the Ca/P ratio did not vary based on the depth from the tidemark (Appendix A), despite the fact that both Ca and P showed depth-dependent behaviors. This reaffirms the consensus in the literature that calcium and phosphorous are co-dependent.

A bone mineralization increase over the depth-range of 50–250 μm (from the tideline) was observed in calves and cattle. Considering the main constituents of hydroxyapatite (i.e., calcium and phosphorus), a similar increase of the Ca and P contents over the same range was expected, which was met in the EDX analysis. Small differences could have been due to a three-dimensional partial volume effect in the mineralization measurement using the micro-CT system.

Limitations of the study include the small sample size, which restricts the statistical power of our analyses to detect subtle changes. Likewise, our work describes phenotypes of bovine subchondral bone at only two ages (3 and 18 months) in timeframes that are inherently immune to OA, which might challenge its relevance to OA. On the other hand, this sampling allowed us to assess purely maturation-related transformations without potential interactions of other risk factors of OA.

## 4. Materials and Methods

### 4.1. Biopsy and Preparation of Bovine Bone Specimens

Eighteen bone specimens from three male calves (three months of age) and three male cattle (18 months of age) were obtained at the day of slaughter at a slaughterhouse. For exposure of the joint surface, the surrounding soft tissue was removed. Bone cylinders were punched out at an inner diameter of 6 mm, and stored in phosphate buffered saline (PBS) at + 5 °C (Figure 1A,B). They were chosen to have a small diameter in order to ensure that a high-resolution (3.5 µm) scanning of the bone microstructure was possible. The location of each set of cylinders was standardized using a template grid, taking into account the normalized size of each condyle, which in turn ensured that the cylinders were extracted from anatomically-similar areas on the medial condyle of femurs. Consequently, K1, K2, and K3 cylinders were used for micro-CT analysis, histology, and EDX spectroscopy, respectively.

### 4.2. Histomorphometry Analysis

Masson–Goldner staining was performed to identify the different layers within the subchondral zone. Undecalcified samples were fixed in 4% neutral buffered formalin for one week, washed repeatedly, dehydrated in ascending concentrations of ethanol (70–100%; each concentration for 3 days), and embedded in methyl methacrylate (MMA; Technovit 9100 New, Heraeus Kulzer GmbH, Hanau, Germany). In addition, 5 µm sections were cut, and after removal of the MMA, the sections were stained with the Masson Trichrome technique. Two histological sections of each sample were analyzed to obtain six sections within each group (calves and cattle). After digital photography at a four-fold magnification, layer thicknesses were measured using ImageJ software (version 1,52a, NIH, Bethesda, MD, USA). The measurements were conducted at 20 fixed spots within the image to obtain a mean value of each zone.

### 4.3. Micro-Computed Tomography (Micro-CT)

For micro-CT Analysis, the samples were processed using a sander and polisher to obtain squares (4 × 4 × 4 mm). They were washed repeatedly with 70% ethanol to remove potential artifacts. A high-definition micro-CT system (micro-CT 35, SCANCO Medical AG, Wangen-Brüttisellen, Switzerland) was used in accordance with established guidelines [42]. At a given isotropic voxel resolution of 3.5 µm, 600 slices were obtained using a source voltage of 70 kVp and an intensity of 114 µA. The automated setting by the manufacturer (lower threshold: 461 mg HA/cm³, upper threshold: 3000 mg HA/cm³, Sigma: 0.8, Support: 1) was selected for our examination. A volume of interest (VOI) of 2000 (length) × 700 (height) × 2000 (width) µm was selected for micro-CT measurements of the subarticular spongiosa. The VOI was placed in the center of the 4 mm^3^ cube, 100 µm below the subchondral endplate, in order to ensure consistency. The measurement of the SCZ’s individual layers was narrowed down to a 700 × 100 × 700 µm cuboid. For the calcified cartilage, this was placed at a depth of 50 µm below the interface of the CC and articular cartilage. To define the region of the subchondral bone plate, the calculated mean value of the CC thickness was added to its two-fold standard deviation. Hence, a VOI of the same size (700 × 100 × 700 µm) was placed 300 µm deep in the bone structure of the calf specimens, and at 435 µm in the cattle.

### 4.4. Representation of Microchannels

A negative image was generated by replacing the values of the upper and lower threshold, which facilitated the identification of cavity structure within the subchondral zone. Thus, the lower threshold was set at −500 mg HA/cm^3^ while the upper threshold at 520 mg HA/cm^3^ (Gauss filter unaltered). 

### 4.5. Mineralization Measurements

Focal analysis of the mineralized matrix was also performed. A frontal layer of the micro-CT scan was analyzed from each sample (Appendix A for details).

### 4.6. Energy Dispersive X-ray Spectroscopy (EDX)

EDX was carried out using an EVO LS 15 electron microscope (Carl Zeiss AG, Oberkochen, Germany). The samples were sputter-coated beforehand with gold for 40 s at a current of 40 mA. The scanning electron microscopy (SEM) images were obtained at 1000-fold magnification, a focus distance of 7 mm and a voltage of 20 keV. The measurement area was a 20 × 20 µm square, while the signal intensity spectrum was recorded for 30 s. Starting from the calcified cartilage, measurements were taken every 50 µm within the first 200 µm of depth, and every 100 µm from 200 to 700 µm. Three different locations were analyzed in each area to obtain a mean value. 

### 4.7. Statistical Analysis

Statistical evaluation was carried out using SPSS package (version 25.0, IBM SPSS, Armonk, NY, USA). One-way analysis of variance (ANOVA) was conducted for the pooled data of layer-by-layer micro-CT analysis. Gaussian distribution was ascertained through the Kolmogorov–Smirnov test and Levene’s test was used for homogeneity of variance. Where there was homogeneity of the variants, the Bonferroni test was consulted for comparisons in pairs. When inhomogeneity was observed, the Dunnett T3 test was used as a post hoc analysis. Where the Gaussian distribution and homogeneity of variance could not be assumed, Kruskal–Wallis one-way ANOVA was conducted with the inclusion of a pairwise post hoc test. To test the hypothesis that difference in sample type (age) and region of the subchondral zone (CC vs. SCBP) leads to different data distribution, Friedman’s two-way ANOVA by ranks was carried out. Statistical significance was set at alpha = 0.05. The correlation between distance from the tidemark and EDX-measured chemical contents/ratios was calculated using Pearson’s correlation coefficient. The best linear fit was plotted, with simultaneous ANOVA test to calculate *p*-values. Throughout the manuscript, data are given as mean ± standard deviation. The level of significance was designated as follows: no significance (NS), *p* < 0.05 (*), *p* < 0.01 (**), and *p* < 0.001 (***).

## 5. Conclusions

In summary, the age-related changes of the healthy bovine bone are mostly governed by an increase in bone volume density induced by thickening of the trabeculae. On the other hand, age demonstrates less influence on the number of trabeculae and their connective network structure compared to the distance from the tidemark. The steep mineralization profile of the calcified cartilage as well as its chemical composition gradient make this layer a potential focal point for future studies. The age-related developmental changes in the SCB are very similar to those of early OA scenarios. As OA is virtually absent in young individuals, the changes in middle-aged healthy bones might include OA-permitting factors. Thus, studying this phase may help to find preventive measures against these changes and consequently against OA. 

## Figures and Tables

**Figure 1 ijms-20-00770-f001:**
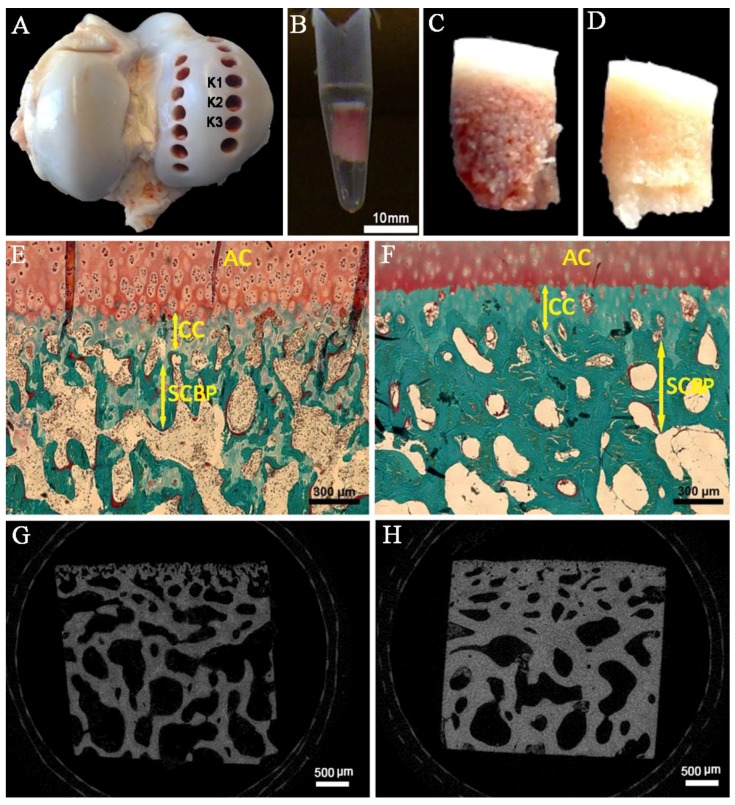
Bone cylinders were punched out from the medial condyle of the distal femur using a bone trephine drill (6.00 mm inner diameter). Cylinders extracted from K1 site on the medial condyle were used for micro-CT analysis, while K2 and K3 samples were selected to perform histomorphometry and EDX evaluations, respectively (**A**). The samples were immediately stored in PBS after the extraction (**B**). Osteochondral biopsies are shown for calf (**C**) and cattle (**D**). Histological sections following Masson–Goldner staining are illustrated for calves (**E**) and cattle (**F**). “AC”, “CC”, and “SCBP” denote articular cartilage, calcified cartilage, and subchondral bone plate, respectively. The microCT image reconstructions are shown in the bottom row. Calves (**G**) demonstrated a finer and thinner trabecular pattern with many small cavities in comparison with cattle (**H**).

**Figure 2 ijms-20-00770-f002:**
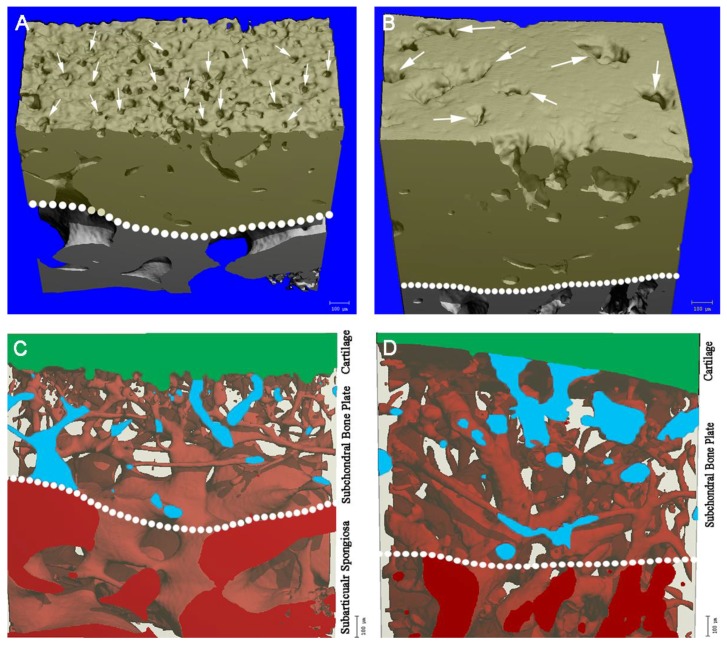
3D-reconstructed images of the subchondral zone in calves (**A**) and cattle (**B**) reveals that the microchannels reach to the most superficial border of the sub-articular junction. Corresponding “negative” images of the exact cross-sections of calves (**C**) and cattle (**D**) were generated by inverting the upper and lower threshold values. Thus, the illustrated red areas are composed of the non-osseous tissues that provide a spatial representation of cavities and the connective microstructure.

**Figure 3 ijms-20-00770-f003:**
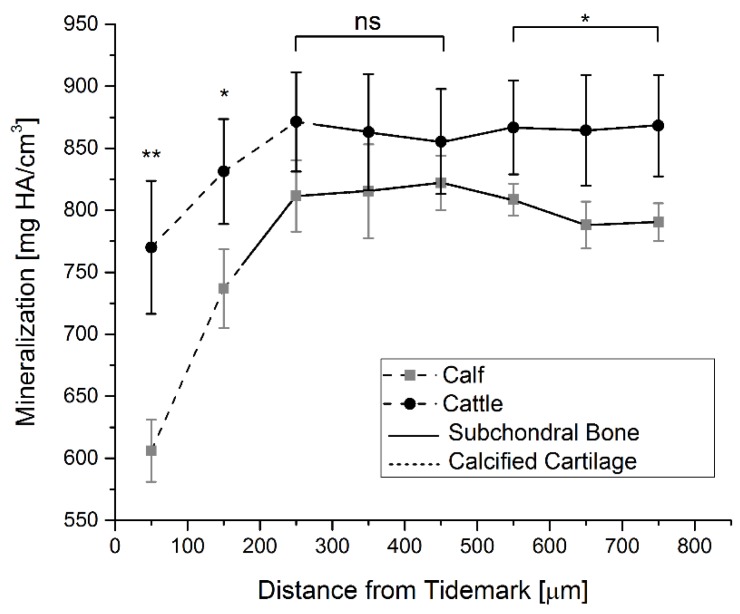
Age/depth-related mineralization of the subchondral bone. Compared to calves, cattle showed higher mineralization values at all distances from the tidemark. The differences were significant at initial (50–150 µm) and far down (550–750 µm) distances from the tidemark, while in the midsection of the SCZ non-significant differences between the age groups were observed. (** *p* < 0.01, * *p* < 0.05) Close to the tidemark, mineralization was significantly lower compared to the region of the SCBP.

**Figure 4 ijms-20-00770-f004:**
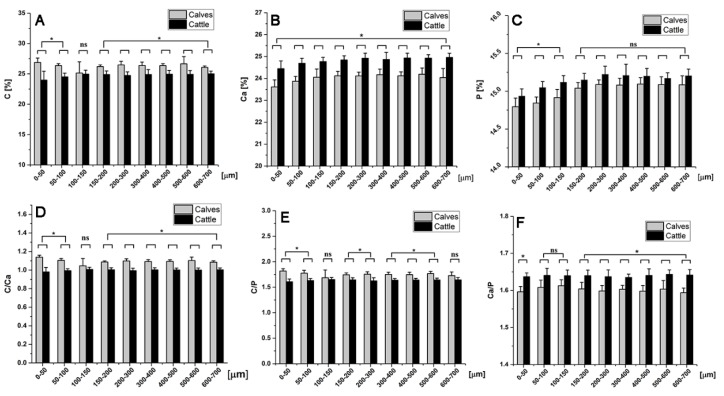
Chemical composition of the SCZ was analyzed via energy-dispersive X-ray spectroscopy (EDX) at different depths. Starting from the tidemark towards the deeper SCBP, carbon, calcium, and phosphorous contents (**A**–**C**), as well as carbon/calcium, carbon/phosphorous, and calcium/phosphorous ratios (**D**–**F**) were measured. * *p* < 0.05.

**Table 1 ijms-20-00770-t001:** Quantitative micro-CT analysis of the subchondral trabecular bone at the medial condyle of distal femur.

Microstructural Parameters of the Subchondral Trabecular Bone	Calves	Cattles	Kruskal–Wallis ANOVA
BV/TV (Bone volume fraction) (%)	48 ± 4	74 ± 4	*p* = 0.049
Tb.Sp. (µm)	45 ± 6	29 ± 2	*p* = 0.046
Tb.N. (1/mm)	11.6 ± 0.8	9 ± 1.5	*p* = 0.049
Conn.D. (1/mm^3^)	3835 ± 1641	1479 ± 292	*p* = 0.049
DA (Degree of anisotropy)	1.14 ± 0.04	1.28 ± 0.12	*p* = 0.275
Tb.Th. (µm)	41 ± 3	85 ± 20	*p* = 0.049

**Table 2 ijms-20-00770-t002:** Quantitative micro-CT analysis of the individual layers of the subchondral bone; One-way ANOVA was performed to evaluate the effect of age (*p*_age_), as well the distance from tidemark (*p*_depth_) on bone indices (* *p* < 0.05, ** *p* < 0.01, *** *p* < 0.001). Mean ± standard deviations (SD) are reported.

Sample Type (a)	Depth	BV/TV (%) Mean ± SD	Tb.Sp. (µm) Mean ± SD	Tb.N. (1/mm) Mean ± SD	Conn.D (1/mm^3^) Mean ± SD	DA Mean ± SD	Tb.Th. (µm) Mean ± SD
Calf	CC	56 ± 4	24 ± 6	18.3 ± 2.2	5115 ± 1783	1.32 ± 0.05	32 ± 1
Cattle	CC	79 ± 11	10 ± 6	16.4 ± 3.0	3315 ± 265	2.17 ± 0.85	52 ± 17
Calf	SCBP	48 ± 3	41 ± 6	12.1 ± 0.9	1782 ± 1085	2.01 ± 0.12	41.5 ± 0
Cattle	SCBP	77 ± 11	15 ± 9	12.7 ± 0.5	995 ± 522	3.05 ± 0.6	64 ± 9
Friedman’s 2-way ANOVA by ranks	*p* *	0.042 *	0.042 *	0.042 *	0.029 *	0.042 *	0.072
One-Way ANOVA	*p_age_*	0.0002 ***	0.0049 **	0.73	0.134	0.029 *	0.006
	*p* _depth_	0.634	0.15	0.001 **	0.027 *	0.08	0.24

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
