# Peer review of "Developmental Transformation and Reduction of Connective Cavities within the Subchondral Bone"

_ijms, 2019, doi:10.3390/ijms20030770_

Round 1
Reviewer 1 Report
International Journal of Molecular Sciences – MDPI
Developmental Transformation and Reduction of Connective Cavities within the Subchondral Bone
Review
Dear Author
The article gives a very interesting new insight in the possible occurrence of osteoarthritic changes in the cartilage, respectively joints. There is special interest in the conduits in the CC region which are more and thinner in the calves specimen which have also a more fine trabecular structure. There are large co-continuous trabecular cavities with small conduits within the CC region and in cattles the trabeculae are thicker, there is more compact bone with only a few conduits into the CC region. Bone Volume density and trabecular thickness increased in age. Even the chemical changes between the different zones are investigated.
For me the presented microstructural changes were very interesting to read and I think there can be a great potential in further studies especially in exploring the “conduits” and what will be going on if osteoporosis is on the way in older specimen – with microscopic and chemical changes in bone density – does the zone of the calcified cartilage gets bigger? or smaller? – preparing the break-down of the cartilage and how this can be influenced in the future.
The article is smooth to read, interesting and and even understandable for a “simple” trauma surgeon – nethertheless it needs a second round to get more or less everything. Especially the presentation of the micro channels is well done ( coloured – fig 2 ). Also the age/depth related mineralisation is presented in a great way. The special investigation concerning carbon, calcium and phosphorus contents is interesting for further age related studies ( the supplementary Table S2 is not visible for me ! ).
Concerning line 166/7 – I do not no what DMM surgery means – Sorry.
If lower subchondral bone thickness in young cartilage is really protective stays questionable. For me the higher number of conduits is the more interesting part of the article which could in my opinion lead to better nutrition and O2-concentration for the upper cartilage which makes him less vulnerable – but this stays hypothetic. Also the advanced “tidemark” which means a little more explanation is an interesting aspect.
The study-specimen were – contributing to the restriction to a special cartilage-bone area and only two age groups – not jet influenced by osteoarthritic changes – therefore it is technically very sound.
So it stays in all parts interesting for the reader. Also the material and methods section – interestingly placed at the number 4 sounds technically sound.
The English is not sophisticated and should be understandable for most of the collegues.
Author Response
Reviewer I:
Dear Author
The article gives a very interesting new insight in the possible occurrence of osteoarthritic changes in the cartilage, respectively joints. There is special interest in the conduits in the CC region which are more and thinner in the calves’ specimen, which have also a finer trabecular structure. There are large co-continuous trabecular cavities with small conduits within the CC region and in cattle the trabeculae are thicker, there is more compact bone with only a few conduits into the CC region. Bone Volume density and trabecular thickness increased in age. Even the chemical changes between the different zones are investigated.
General response:
Thank you very much for taking the time to assess our manuscript. We appreciate your overall positive evaluation, as well as your insightful suggestions. We have revised the manuscript according to your comments.
1- For me the presented microstructural changes were very interesting to read and I think there can be a great potential in further studies especially in exploring the “conduits” and what will be going on if osteoporosis is on the way in older specimen – with microscopic and chemical changes in bone density – does the zone of the calcified cartilage gets bigger? or smaller? – preparing the break-down of the cartilage and how this can be influenced in the future.
Response
We want to thank the reviewer for pointing out the great potential of our data as a basis for future understanding of the subchondral zone.
2- The article is smooth to read, interesting and even understandable for a “simple” trauma surgeon – nevertheless it needs a second round to get more or less everything. Especially the presentation of the micro channels is well done (colored – fig 2). Also the age/depth related mineralization is presented in a great way. The special investigation concerning carbon, calcium and phosphorus contents is interesting for further age related studies (the supplementary Table S2 is not visible for me!).
Response/action:
Again, we appreciate the positive evaluation of our manuscript, and apologize for the inconvenience regarding the supplementary table. We have now copied the supplementary table below, and have uploaded a new version with our revised submission. Hopefully, the file can be viewed properly now.
Supplementary table. Statistical correlation between chemical composition and distance from the tidemark in bovine bone
Chemical Content/Ratio | Sample Type | Depth | Intercept | Pearson’s r | p |
C [%] | Calves | 0-300 [µm] | 26.39 | -0.18 | 0.77 |
300-700 [µm] | 26.67472 | -0.32374 | 0.67626 | ||
Cattle | 0-300 [µm] | 24.19875 | 0.71078 | 0.17839 | |
300-700 [µm] | 24.79778 | 0.80013 | 0.19987 | ||
Ca [%] | Calves | 0-300 [µm] | 23.67 | 0.88 | * 0.045 |
300-700 [µm] | 24.29979 | -0.62918 | 0.37082 | ||
Cattle | 0-300 [µm] | 24.48015 | 0.93823 | * 0.01826 | |
300-700 [µm] | 24.78601 | 0.89774 | 0.10226 | ||
P [%] | Calves | 0-300 [µm] | 14.75 | 0.98 | ** 0.003 |
300-700 [µm] | 15.08543 | 0.03824 | 0.96176 | ||
Cattle | 0-300 [µm] | 14.93755 | 0.97117 | ** 0.00585 | |
300-700 [µm] | 15.21708 | -0.30443 | 0.69557 | ||
C/Ca | Calves | 0-300 [µm] | 1.11 | -0.41 | 0.49 |
300-700 [µm] | 1.09782 | -0.17946 | 0.82054 | ||
Cattle | 0-300 [µm] | 0.98827 | 0.46032 | 0.43532 | |
300-700 [µm] | 1.00048 | 0.11789 | 0.88211 | ||
C/P | Calves | 0-300 [µm] | 1.78 | -0.47 | 0.43 |
300-700 [µm] | 1.76825 | -0.32823 | 0.67177 | ||
Cattle | 0-300 [µm] | 1.61991 | 0.41784 | 0.48391 | |
300-700 [µm] | 1.62959 | 0.98665 | * 0.01335 | ||
Ca/P | Calves | 0-300 [µm] | 1.6 | -0.06 | 0.92 |
300-700 [µm] | 1.61082 | -0.62717 | 0.37283 | ||
Cattle | 0-300 [µm] | 1.63911 | -0.10656 | 0.86458 | |
300-700 [µm] | 1.62882 | 0.81477 | 0.18523 |
3- Concerning line 166/7 – I do not know what DMM surgery means – Sorry.
Response/action:
We acknowledge that we should have clarified the abbreviation upon its first use. DMM stands for “destabilization of the medial meniscus”; a popular model to induce osteoarthritis in rats and mice. The terminology has now been added to the Discussion (page 8, line 194), as well as to the Abbreviations (p. 11, l. 356).
4- If lower subchondral bone thickness in young cartilage is really protective stays questionable. For me the higher number of conduits is the more interesting part of the article which could in my opinion lead to better nutrition and O2-concentration for the upper cartilage which makes him less vulnerable – but this stays hypothetic. Also the advanced “tidemark” which means a little more explanation is an interesting aspect.
Response:
We agree with the reviewer, that our results so far mainly allow the formulation of hypotheses that can be used as foundation for future studies. Indeed, the possible nutrition and O2 supply to the cartilage is an interesting aspect and we have now included an additional sentence about possible nutritional and oxygen supply through these channels into the discussion (p. 8, l. 220-1).
5- The study-specimen were – contributing to the restriction to a special cartilage-bone area and only two age groups – not jet influenced by osteoarthritic changes – therefore it is technically very sound.
So it stays in all parts interesting for the reader. Also the material and methods section – interestingly placed at the number 4 sounds technically sound.
The English is not sophisticated and should be understandable for most of the colleagues.
Response:
Thank you very much for appreciating our manuscript, and the helpful comments to further improve its quality.
Reviewer 2 Report
In this study, Taheri et al., set out to characterize the effects of ageing on subchondral bone. This was done to understand the potential changes within subchondral bone typically associated with early and late osteoarthritis. The study is well done overall, and it is scientifically sound. The writing has been well done as well. I think that the article merits further consideration, however there are a few points which must be addressed.
1) The authors imply (although it was indirectly) that using bovine ageing model of knee joints from 3 month old calves and 18 month old cattle are somehow similar ageing in human knee joints (which they also imply are young and healthy less than 30 years of age and older after 30 years). The problem with using this model to assess changes in ageing on the subchondral bone is that 3 month calves (weight about 60kg) have much less compressive strain on their joints that 18 month cattle (which are about 150 kg). The last time I checked, my weight (at ~40 years old) is almost identical to my weight at 18-20 years of age. There has been no added compressive strain (other than heavy excercise and some sport) to my joints. What I am trying to say here is that the assumption that changes to subchondral bone in humans aged 18-60 is probably not equivalent to changes in bovine model from 3 months - 18 months. I think more details about this is necessary in the introduction to justify your model, and also this must be discussed at the end as well.
2) the difference in trabecular bone thickness is quite striking between the two age groups. in figure 1, it would add a lot of vale to stain the sections for osteoblastic activity (ALP), osteoclast activity (TRAP) and vascularization (CD34). This will potentially help address part of the mechanism at play.
Author Response
Reviewer II:
Comments and Suggestions for Authors
In this study, Taheri et al., set out to characterize the effects of ageing on subchondral bone. This was done to understand the potential changes within subchondral bone typically associated with early and late osteoarthritis. The study is well done overall, and it is scientifically sound. The writing has been well done as well. I think that the article merits further consideration, however there are a few points which must be addressed.
General response
We would like to thank the reviewer for his/her positive overall evaluation of our study. The manuscript has been certainly benefited from your insightful comments.
1) The authors imply (although it was indirectly) that using bovine ageing model of knee joints from 3 month old calves and 18 month old cattle are somehow similar ageing in human knee joints (which they also imply are young and healthy less than 30 years of age and older after 30 years). The problem with using this model to assess changes in ageing on the subchondral bone is that 3 month calves (weight about 60kg) have much less compressive strain on their joints that 18 month cattle (which are about 150 kg). The last time I checked, my weight (at ~40 years old) is almost identical to my weight at 18-20 years of age. There has been no added compressive strain (other than heavy exercise and some sport) to my joints. What I am trying to say here is that the assumption that changes to subchondral bone in humans aged 18-60 is probably not equivalent to changes in bovine model from 3 months - 18 months. I think more details about this is necessary in the introduction to justify your model, and also this must be discussed at the end as well.
Response
We thank the reviewer for raising this valid point. We have added additional paragraphs into Introduction and Discussion to justify the model and discuss the results.
Unquestionably, joint-compressive load is a major contributing factor in shaping bone architecture, and as mentioned by the reviewer, there is a 2.5-fold difference between the weights of 3 month-old calves and 18 month-old cattle. Nonetheless, in order to determine the accurate equivalent age between different species (e.g. cattle and human), one needs to take into account the lifespan differences, as well. Cattle have a natural lifespan of 15-23 years, while the average lifespan of human is reportedly 75-80 years. Hence, 3 month-old calves and 18 month-old cattle, which are representative of early age and young adulthood are equivalent to 3-6 year-old preschoolers and 16-18 year-old young adults in human, respectively.
The weight of 3-6 year-old preschoolers is in the range of 17.3-23.0 kg, with a mean value of ∼ 20 kg. On the other hand, 16-18 year-old young adults weigh 60.8-67 kg on average, which accounts to ∼ 300% weight difference between the two groups. When compared to the 250% weight difference of 3 month-old calves and 18 month-old cattle, the implication does not seem far-fetched. In fact, we tried to be really careful with our selection of the age-classes, since we wanted to address how pure maturation could change the subchondral bone microstructure. Therefore, we selected two pre-OA groups that represent young age and early adulthood. Admittedly, a more scientifically-sound approach would have been to compare the mean ratio of the body mass index (BMI) among the two species. However, there is a lack of data for the BMI values of animals in the literature.
Action
In line with the reviewer’s suggestions, this issue is now addressed in the Introduction, and a detailed explanation is provided in the Discussion. The last paragraph of the Introduction reads now:
“To uncover possible microstructural changes of the SCZ at a potentially interesting time frame at the border between early age and young adulthood, we analyzed the chemical composition, histology, and bone microstructure of healthy subchondral bone in 3-month-old calves and 18-month-old cattle. Even though inherent differences between the ageing models of bovine and human might exist, we will discuss that the presented model could be valuable in addressing key microstructural changes in the maturation phase of the subchondral bone.”
Likewise, the Discussion now reads:
“There might be obvious interspecies differences between the bone maturation of bovine and human bones, even though comparative genomics mapping of cows and humans shows that over 80% of the examined cattle genes had human orthologs and the two genomes had at least 105 conserved chromosomal segments in common [18]. Nevertheless, the weight of 18-month-old cattle (~150 kg) is 2.5 times higher than that of calves (~ 57 kg) [19,20]. As joint-compressive load is considered a major contributing factor in shaping bone architecture [21], this might raise the question of whether the bovine ageing model can be compared to ageing in human. However, in order to determine the proper correspondence between different age-groups of cattle and human, the lifespan differences need to be addressed, as well. Cattle have a natural lifespan of 15-23 years [22,23], while the natural lifespan of human is reportedly 75-80 years [24]. Hence, 3 month-old calves and 18 month-old cattle, which are representative of early age and young adulthood are equivalent to 3-6 year-old preschoolers and 16-18 year-old young adults in human, respectively [18,22]. According to Vorwerg et al., the weight of 3-6 year-old preschoolers is in the range of 17.3-23.0 kg, with a mean value of ∼ 20 kg [25]. On the other hand, 16-18 year-old young adults weigh 60.8-67 kg [26,27], which accounts to ∼ 300% weight difference between the two groups. Taking into consideration the 250% weight difference between 3 month-old calves and 18 month-old cattle, it can be concluded that our sampling allows comparable maturation-based observations. Therefore, two pre-OA groups that represent young age and early adulthood were selected.” (p. 7, l. 166-183).
Added bibliography:
18. Band, M.R.; Larson, J.H.; Rebeiz, M.; Green, C.A.; Heyen, D.W.; Donovan, J.; Windish, R.; Steining, C.; Mahyuddin, P.; Womack, J.E.; Lewin, H.A. An ordered comparative map of the cattle and human genomes. Genome Res. 2000, 10, 1359–68, doi:10.1101/GR.145900.
19. Mummed, Y.Y. Correlation between milk suckled and growth of calves of ogaden cattle at one, three and six months of age, east Ethiopia. Springerplus 2013, 2, 302, doi:10.1186/2193-1801-2-302.
20. Fall, A.; Diop, M.; Sandford, J.; Wissocq, Y.J.; Durkin, J.W.; Trail, J.C.M. Evaluation of the productivities of Djallonke sheep and N’Dama cattle at the Centre de Recherches Zootechniques, Kolda, Senegal. 1982.
21. Ryan, T.M.; Shaw, C.N. Trabecular bone microstructure scales allometrically in the primate humerus and femur. Proc. R. Soc. B Biol. Sci. 2013, 280, 20130172–20130172, doi:10.1098/rspb.2013.0172.
22. Burgstaller, J.P.; Brem, G. Aging of Cloned Animals: A Mini-Review. Gerontology 2017, 63, 417–425, doi:10.1159/000452444.
23. Costagliola, A.; Wojcik, S.; Pagano, T.B.; De Biase, D.; Russo, V.; Iovane, V.; Grieco, E.; Papparella, S.; Paciello, O. Age-Related Changes in Skeletal Muscle of Cattle. Vet. Pathol. 2016, 53, 436–446, doi:10.1177/0300985815624495.
24. Dong, X.; Milholland, B.; Vijg, J. Evidence for a limit to human lifespan. Nature 2016, 538, 257–259, doi:10.1038/nature19793.
25. Vorwerg, Y.; Petroff, D.; Kiess, W.; Blüher, S. Physical Activity in 3–6 Year Old Children Measured by SenseWear Pro®: Direct Accelerometry in the Course of the Week and Relation to Weight Status, Media Consumption, and Socioeconomic Factors. PLoS One 2013, 8, e60619, doi:10.1371/journal.pone.0060619.
26. Diehr, P.; O’Meara, E.S.; Fitzpatrick, A.; Newman, A.B.; Kuller, L.; Burke, G. Weight, mortality, years of healthy life, and active life expectancy in older adults. J. Am. Geriatr. Soc. 2008, 56, 76–83, doi:10.1111/j.1532-5415.2007.01500.x.
27. Chiu, H.-C.; Chang, H.-Y.; Mau, L.-W.; Lee, T.-K.; Liu, H.-W. Height, Weight, and Body Mass Index of Elderly Persons in Taiwan. Journals Gerontol. Ser. A Biol. Sci. Med. Sci. 2000, 55, M684–M690, doi:10.1093/gerona/55.11.M684.
2) the difference in trabecular bone thickness is quite striking between the two age groups. in figure 1, it would add a lot of value to stain the sections for osteoblastic activity (ALP), osteoclast activity (TRAP) and vascularization (CD34). This will potentially help address part of the mechanism at play.
Response
We totally agree with the reviewer that presenting a variety of immunohistochemistry staining would be valuable for understanding the cellular and molecular mechanisms behind the observed changes. However, this would have come at the expense of shifting the focus of the study to a cellular and molecular biology level. This was neither our intention, nor was manageable within the project’s timeframe, especially as commercially available antibodies are often not tested for bovine tissue. Instead, we tried to explore the differences in the microstructure, chemical composition, and mineralization of the young vs. early adult bovine bone.
Having said that, it might be noteworthy to mention that we’re currently translating our results to healthy vs. osteoarthritic human hip joints, where a variety of immunohistochemistry stainings such as CD31, TRAP, von Willebrand factor (VWF), Calcitonin receptor (CT), Collagen I, etc. are being conducted. As the reviewer suggests, these results can hopefully shed light on the molecular mechanism at play.
Round 2
Reviewer 2 Report
no further comments